# The Research Status, Potential Hazards and Toxicological Mechanisms of Fluoroquinolone Antibiotics in the Environment

**DOI:** 10.3390/antibiotics12061058

**Published:** 2023-06-15

**Authors:** Jia Du, Qinghua Liu, Ying Pan, Shaodan Xu, Huanxuan Li, Junhong Tang

**Affiliations:** 1College of Materials and Environmental Engineering, Hangzhou Dianzi University, Hangzhou 310018, China; panying@hdu.edu.cn (Y.P.); xusd@hdu.edu.cn (S.X.); hxlee@hdu.edu.cn (H.L.); tangjunhong@hdu.edu.cn (J.T.); 2Suzhou Fishseeds Biotechnology Co., Ltd., Suzhou 215138, China; qhliu.visitingp@xjtlu.edu.cn; 3Hongze Fishseeds Biotechnology Co., Ltd., Huaian 223125, China; 4Wisdom Lake Academy of Pharmacy, Xi’an Jiaotong-Liverpool University, Suzhou 215123, China

**Keywords:** fluoroquinolone antibiotics, ecotoxicity, residues, human consumption, oxidative stress-related mechanism, antibiotic resistance

## Abstract

Fluoroquinolone antibiotics are widely used in human and veterinary medicine and are ubiquitous in the environment worldwide. This paper recapitulates the occurrence, fate, and ecotoxicity of fluoroquinolone antibiotics in various environmental media. The toxicity effect is reviewed based on in vitro and in vivo experiments referring to many organisms, such as microorganisms, cells, higher plants, and land and aquatic animals. Furthermore, a comparison of the various toxicology mechanisms of fluoroquinolone antibiotic residues on environmental organisms is made. This study identifies gaps in the investigation of the toxic effects of fluoroquinolone antibiotics and mixtures of multiple fluoroquinolone antibiotics on target and nontarget organisms. The study of the process of natural transformation toward drug-resistant bacteria is also recognized as a knowledge gap. This review also details the combined toxicity effect of fluoroquinolone antibiotics and other chemicals on organisms and the adsorption capacity in various environmental matrices, and the scarcity of data on the ecological toxicology evaluation system of fluoroquinolone antibiotics is identified. The present study entails a critical review of the literature providing guidelines for the government to control the discharge of pollutants into the environment and formulate policy coordination. Future study work should focus on developing a standardized research methodology for fluoroquinolone antibiotics to guide enterprises in the design and production of drugs with high environmental biocompatibility.

## 1. Introduction

Antibiotics are widely used in medicines for treating bacterial infections in both people and animals due to their biologically active antibacterial, antifungal, and antiparasitic activities [1,2]. Fluoroquinolone (FQ) antibiotics are the most commonly used antibiotics in animal agriculture, animal aquaculture, and human medicine, owing to their broad antimicrobial activity in the treatment of the urinary, gastrointestinal, abdominal, and respiratory tracts [3]. FQ antibiotics are popular in antimalarial drug and synthetic antibacterial drug animal agriculture and aquaculture [4]. The first- and second-generation FQ antibiotics entered the market one after another, including nalidixic acid, ciprofloxacin, lomefloxacin, enrofloxacin, ofloxacin, and norfloxacin [5,6]. Byproducts and effluent have dramatically increased in recent years with the production of FQ antibiotics. FQ antibiotics enter the environment by different routes, including processing, use, and disposal [7]. The effluent from municipal plumbing and farm water is discharged without treatment into local waterways, which also increases the concentration of antibiotics in the water. FQ antibiotics are considered one of the major contaminant antibiotics due to the difficulty of removing them from wastewater [7]. The annual production of FQ antibiotics is estimated to be 1520 tons globally, and 70% of annual production occurs in the United States and the European Union [8]. China and India are the major food suppliers in Asia and have used antibiotics as growth promoters [9]. Human travel, animal migration, and the import and export of food have also increased the movement of antibiotics worldwide [10]. FQ antibiotics are often found in wastewater, biosolids, soil, and sediments worldwide [11,12]. Humans have raised concerns about the fate and potential biological toxicity of FQ antibiotics [13]. According to a report from the World Health Organization, FQ antibiotics pose a severe threat to human health and are abundant in the environment [14].

FQ antibiotics are considered toxic to bacteria, mammals, cells, and plants in a wide range of concentrations [14,15]. Scholars usually analyzed the toxicity of FQ antibiotics to organisms by exposure routes, including oral, sorption, and injection routes. Kumar et al. (2012) found that the toxicity of FQ antibiotics to plants mostly depends on the sorption of FQ antibiotics in soil [16]. The potential risk to human health is rising by consuming foods containing FQ antibiotic residues [17]. In addition, the sorption activity and antibacterial responses of FQ antibiotics are mainly dependent on their physiological characteristics and environmental compatibility [18,19]. FQ antibiotics with different structures of piperazinyl and substituent may occur in different speciation when discharged into the environment [20]. The complex physiological characteristics of FQ antibiotic mixtures may affect their toxic effects on organisms in different environments. The single and joint toxic hazards of FQ antibiotics and other pollutants on organisms were determined in vitro and in vivo. Systematic experimental methods can accurately analyze the toxic effects of compounds. However, the bioaccumulation and biomagnification of FQ antibiotics interacting with their structure in the environment is poorly understood. Understanding the toxicity impact of FQ antibiotic action is necessary to assess the environmental risk. In addition, understanding the toxicity mechanisms of FQ antibiotics might help to better analyze the spread of resistance. Previous research mostly studied the actual hazard of single antibiotics. Therefore, this systematic review summarized the available coexposure toxicity data of FQ antibiotics with other contaminants.

In recent years, reviews about the recent trends, applications, and toxicity of multiple antibiotics have rapidly increased in quantity. However, an overview of FQ antibiotics on organisms with a wide range of classifications (microorganisms, cells, lower or higher plants, and terrestrial or aquatic animals) in both in vivo and in vitro experiments has yet to be described, to our knowledge. This research aims to provide a comprehensive literature overview of the single and joint toxicity effects of FQ antibiotics on organisms and the underlying toxicity mechanisms based on in vivo culture cell and in vitro experimental organism studies. The core database Web of Science was searched on 10 April 2023, for articles in the literature that discuss FQ antibiotic toxic hazards. “Mechanism” or “in vivo” and “in vivo” or “antibiotics toxicology” or “fluoroquinolones” were used as a “topic”. The selected document types were articles and review articles. English was selected as the language. After an initial evaluation of the content, the annual and accumulated publications from 1999 to 2023 focused on the toxic hazard of FQ antibiotic exposure (as shown in Figure 1), both in vivo and in vitro toxicity. After a systematic analysis, 145 published articles were obtained. Aside from providing an in-depth look at the development and risks of FQ antibiotics when they enter the environment, the present study entails a critical review of the literature and motivates researchers to carry out further research.

## 2. Source and Fate of FQ Antibiotics in the Environment

FQ antibiotics enter the environment via biosolids, animal manure, antibiotic manufacturing, and wastewater from households, hospitals, and industries [21,22,23,24]. FQ antibiotics are used in animal agriculture, including in the European Union, North America, Australia, and Asia [25]. For example, enrofloxacin and danofloxacin are used for the treatment of bovine respiratory disease in Canada [26]. Enrofloxacin and danofloxacin are used to treat Escherichia coli infections in swine in the United States [27]. Sarafloxacin is only authorized for use in turkeys and chickens in the United States [28]. FQ antibiotics are discharged from animal and human body residues in manure form or as metabolites, especially in livestock animals [29]. Researchers detected quite high concentrations of ciprofloxacin, enrofloxacin, and norfloxacin in chicken droppings in China [8,30,31]. The feces of livestock and poultry were processed and modified into fertilizer or sludge to apply to the land for agriculture, which was the primary source of pollution in agricultural fields [32]. Through the process of rainwater rewash or landmark runoff, these metabolites of FQ antibiotics enter the groundwater, rivers, lakes, and seas [33].

Aquaculture leads to a lot of FQ antibiotics inflowing into the surrounding water and land worldwide. In commercial aquaculture-intensive regions, FQ antibiotics are sprayed directly into the water for treating and preventing the bacterial disease of aquatic animals [34]. Enrofloxacin, norfloxacin, ciprofloxacin, and oxalinic acid are commonly used in commercial aquaculture, including fish, mollusks, crustaceans, and aquatic plants [34,35]. Large quantities of the FQ antibiotics used for aquaculture have been found in river water, lake water, and sea water, which are important sources of FQ antibiotics in the water environment. Globally, human consumers in some countries have no solution for consuming meat, livestock, or aquaculture products from animals raised on feed containing antibiotics. FQ antibiotic residues or their metabolites are discharged from the human body and enter a city’s sewage disposal system from households. FQ antibiotics are medicines used to treat diseases caused by bacterial infections in humans, such as respiratory, urinary tract, intestinal, and abdominal infections. The source of the FQ antibiotics in human pharmaceuticals and consumer goods is released into the water environment via domestic sewage, hospital wastewater, and industrial wastewater [36]. Most of these places do not have water treatment systems, so waste is directly discharged into urban sewage systems. FQ antibiotics are often detected in hospital wastewater, and norfloxacin is the most frequently detected antibiotic among these antibiotics [37]. In hospital sewage, some researchers detected ofloxacin and ciprofloxacin with concentrations ranging from 3 mg/L to 87 mg/L in European countries [38,39]. In addition, the pharmaceutical production industry is also the main source of FQ antibiotics for sewage treatment works [40]. Fick et al. (2009) found that the highest concentration of ciprofloxacin (14 mg/L) was detected in the waste water treatment systems of pharmaceutical production as well as near lakes [41]. The high concentrations of FQ accumulated in the treated byproduct sludge from the waste water treatment system. The removal capability of FQ antibiotics in wastewater samples was generally 60% [42]. Within the digested sludge, norfloxacin, ciprofloxacin, enrofloxacin, fleroxacin, ofloxacin, and lomefloxacin could be found in the residual [43]. Most of the residual of FQ antibiotics remains in the waste water treatment system and reaches nearby water bodies [44].

The pharma-kinetic and physiochemical properties, climatic conditions, temperature, pH, and light are the main factors determining the occurrence and fate of FQ antibiotics in the environment [45,46]. The polarity and water solubility of antibiotics affect the equilibrium, sorption–desorption, and sequestration processes in the soil environment. The half-life of antibiotics is negatively correlated with their bioaccessibility and bioavailability in the soil environment [47,48]. However, the bioavailability and bioaccessibility of antibiotics are negatively correlated with their existence time in the soil. Jechalke et al. (2014) found that FQ antibiotics could remain in the soil for years [49]. The water solubility and flow properties of FQ antibiotics play an important role in increasing their concentrations and mobility in water environments, including surface water [24,35,50,51], groundwater [52], sewage [37,42] and sediment samples [7,35,53]. Chee-Sanford et al. (2009) found that the water solubility of ciprofloxacin and enrofloxacin was 30 g/L and 130 g/L, respectively, which were considered hydrophilic (exceeding 1 g/L) [54]. Yun et al. (2018) found that photodegradation was an important characteristic affecting the existential state of FQ antibiotics in the aquatic environment [55]. For example, the good water solubility and long half-life characteristics of FQ antibiotics result in high residue concentrations and long-term existence in the environment [19]. Rico et al. (2014) found that enrofloxacin (from 1.4 mg/kg to 2339 mg/kg dry weight) had high binding properties and a long half-life in the river environment [51]. The low biodegradability and absorption characteristics of FQ antibiotics resulted in their residues and metabolites being detected in soil and water sources. The polarity and water solubility of antibiotics control some processes in the solid soil phase, such as equilibrium, absorption–desorption, and sequestration. These processes, together with pH and pore size, are also a function of soil characteristics.

## 3. Toxicity Effect of FQ Antibiotics on Organisms

### 3.1. In Vitro Studies on FQ Antibiotics

The toxic potential of FQ antibiotics was evaluated by using in vitro methods to indicate adverse health effects on mammalian cells (Table 1). Eight fluoroquinolones (norfloxacin, ofloxacin, enoxacin, ciprofloxacin, lomefloxacin, tosufloxacin, sparfloxacin, and gatifloxacin) showed deoxyribonucleic acid (DNA) strand-breaking activities under UV-A irradiation after exposure to mammalian cells [56]. FQs attract attention because of their broad-spectrum antibacterial activity. Ciprofloxacin and moxifloxacin show better in vitro activities than most respiratory tract pathogens. Moxifloxacin and ciprofloxacin protect human respiratory epithelial cells against *Streptococcus pneumoniae*, *Staphylococcus aureus*, *Pseudomonas aeruginosa*, and *Haemophilus influenzae* in vitro [57]. The results suggest that the intracellular accumulation of moxifloxacin and ciprofloxacin plays an important role in the protection of respiratory epithelial cells from the cytotoxic effects of major respiratory tract pathogens. Moxifloxacin and gatifloxacin at a concentration of 1000 µM suppressed primary monkey hepatic gluconeogenesis, which might be related to the clinically relevant dysglycemia regulated by FQ antibiotics in humans [58]. FQ antibiotics resulted in a significant reduction in cultivated (human, dog, mini pig, rat, and marmoset) tendon cell viability [59]. Intrinsic cytotoxic effects were found after human corneal keratocytes and endothelial cells were exposed to one of five fluoroquinolones (ciprofloxacin, gatifloxacin, levofloxacin, moxifloxacin, or ofloxacin) at concentrations of 1 mg/mL, 100 μg/mL, 10 μg/mL, 1 μg/mL, 100 ng/mL, or 10 ng/mL for 15, 30, 60, or 240 min [60]. Matsumoto et al. (2006) found that gatifloxacin was less inhibitory to the processes involved in corneal re-epithelialization [61]. Bai et al. (2015) suggested that levofloxacin had cytotoxic effects on rat annulus fibrosus cells, which was characterized by enhancing apoptosis and reducing cell viability, and indicated a potential toxic effect of FQ antibiotics on rat annulus fibrosus cells [62]. Moxifloxacin and ciprofloxacin upregulated the type III secretion system and did not negate cytotoxic effects in corneal epithelial cell infection models [63]. Marquez-Lazaro et al. (2022) found potential negative effects of sarafloxacin residues on chicken meat, which could affect its nutritional and technological properties [64]. Besifloxacin, gatifloxacin, and moxifloxacin were exposed to human umbilical vein endothelial cells, resulting in a significant loss of cell viability after 24 h of exposure [65]. Clinically relevant concentrations of ciprofloxacin and tetracycline had detrimental impacts on human retinal pigment epithelial cell (ARPE-19 cell) lines in vitro, including the upregulation of genes related to apoptosis, inflammation, and the antioxidant pathways [66].

Some of the susceptibility of bacteria in in vitro studies also indicated the toxic effect of FQ antibiotics, and their relevance to human drug use due to their broad-spectrum bactericidal activity and low toxicity is still under discussion. In the challenges of assessing microbial susceptibility and predicting the clinical response to newer-generation FQs, besifloxacin, gatifloxacin, and moxifloxacin are useful as treatment drugs for eye bacterial infections [68]. In an experiment analyzing the antibiotic susceptibilities of *Parachlamydia acanthamoeba* in Amoebae, Maurin et al. (2002) indicated that the two strains (strain Bn (9) and Hall’s coccus) in *Acanthamoeba polyphaga* were resistant to FQs [69]. The in vitro activity of ciprofloxacin, ofloxacin, levofloxacin, sparfloxacin, and gatifloxacin against multidrug-resistant *Mycobacterium tuberculosis* in Rio de Janeiro, Brazil, also indicated that various FQ antibiotics might be effective therapeutic alternatives in infections caused by *Mycobacterium tuberculosis* [70]. Frean et al. (2003) studied the susceptibility of *Bacillus anthracis* to novel FQ antibiotics [71]. They found that trovafloxacin was very active against Gram-negative and Gram-positive bacteria [71]. Andoh et al. (2004) evaluated the in vitro susceptibility to tetracycline and FQ antibiotics of Japanese isolates of *Coxiella burnetii* for the first time [72]. Murray et al. (1993) found that the ranking of the in vitro activity of nine fluoroquinolone antibiotics against 200 strains of enterococci was sparfloxacin > ciprofloxacin, temafloxacin > ofloxacin > fleroxacin, lomefloxacin, and norfloxacin > enoxacin [73]. Trujillano-Martin et al. (1999) tested the in vitro activities of eight fluoroquinolones against 160 *Brucella melitensis* strains [74]. The most active was sitafloxacin. After a comparison of the bactericidal activity of quinolone antibiotics in a *Mycobacterium fortuitum* model, researchers found that the optimum bactericidal concentrations for the FQs were moxifloxacin, 0.5 mg/L; ciprofloxacin and sparfloxacin, 2 mg/L; and ofloxacin, 8 mg/L [75]. In addition, Koss et al. (2007) found new FQs (levofloxacin, gatifloxacin, and moxifloxacin) that were effective against the normal bacterial flora of the conjunctiva [76]. In a study of the bactericidal activity of FQs against plasmid-mediated QnrA-producing *Escherichia coli*, it was found that ofloxacin and ciprofloxacin had strong bactericidal activity [77]. Identifying the in vitro activity of FQs against *Mycoplasma genitalium* and their bacteriological efficacy for the treatment of *M. genitalium*-positive nongonococcal urethritis in men, Yasuda et al. (2005) found that levofloxacin was less active than gatifloxacin, tosufloxacin, and sparfloxacin [78]. Otherwise, FQs have been monitored closely since they were licensed for veterinary medicine. In an in vitro study of antibiotic resistance in the efficacy of enrofloxacin and marbofloxacin on bacteria isolated from dogs and cats, Muller et al. (2009) found that enrofloxacin had equal or even slightly higher efficacy than marbofloxacin, except regarding *Pseudomonas aeruginosa* [79].

### 3.2. In Vivo Studies on FQ Antibiotics

As shown in Table 2, in vivo studies of FQ antibiotic toxicity involve many species, such as invertebrates and vertebrates. Steward et al. (2004) indicated the efficacies of prophylactic and therapeutic gatifloxacin and moxifloxacin in a BALB/c mouse model of systemic and pneumonic plague [80]. Rizk et al. (2018) showed that enoxacin and diminazene aceturate combination therapy exhibited a potential antibacterial effect when mice were infected with Babesia microti. Their findings provide evidence for enoxacin treatment of clinical tuberculosis caused by *Babesia* spp. [81]. Maitre et al. (2017) found that both levofloxacin and moxifloxacin were active against drug-susceptible tuberculosis. However, moxifloxacin had a better curative effect on susceptible Mycobacterium tuberculosis strains in mice [82]. Shadoud et al. (2015) identified that FQ-resistant mutations in Legionella pneumophila occurred during hospitalization, most likely after fluoroquinolone treatment [83]. FQ antibiotics decreased morbidity and mortality and inhibited Campylobacter in chickens, which could decrease the transmission of Campylobacter via the food chain [84]. FQ antibiotics have been widely used for therapeutic purposes in aquaculture production. Regmi et al. (2007) suggested that these FQs lacked inhibitory effects on cytochrome P-450 3A activities in dogs in vivo and in vitro. Therefore, there is a risk of drug–drug interactions between FQs and P-450 3A activity substrates in clinical situations in dogs [85]. Li et al. (2022) found that paeoniflorin and norfloxacin cotreatment was effective in treating Streptococcus suis infections in mice [86]. Ullah et al. (2022) found that moxifloxacin and gemifloxacin significantly reduced blood glucose levels both in an in vivo rabbit model and in the test subjects of a phase I clinical trial [87]. FQs have different effects on host bile acid metabolism. For example, ciprofloxacin significantly reduced the hydrophobicity index of the bile acid pool of male Wistar rats, reduced secondary bile acids, and increased taurine-conjugated primary bile acids in both the serum and large intestine compared with moxifloxacin [88]. Severe cartilage erosions, synovitis, and joint capsular thickening were identified in foals postnatally treated with enrofloxacin, indicating that in utero exposure to FQs may cause subtle lesions [89]. Tang et al. (2017) found that FQ antibiotics were enriched in the dorsal-epaxial muscle of living puffer fish, thus causing health risks for humans [90]. Exposure to minocycline alone or a combination of minocycline and levofloxacin induced the highest survival rate (70%) in zebrafish, increasing the antimicrobial effects against Elizabethkingia anophelis [91]. Du et al. (2021) also found that enrofloxacin interfered with many signaling pathways related to protein and lipid accumulation in American shad (*Alosa sapidissima*), including the glycolysis/gluconeogenesis and pentose phosphate pathways [92]. Xiao et al. (2018) revealed a relationship between fluoroquinolone structure and neurotoxicity. They found that structural modifications of FQs can change toxicity properties in zebrafish [93].

The toxic effect of FQ antibiotic exposure on higher plants in vivo was also discussed. Jin et al. (2022) found that enrofloxacin, norfloxacin, and levofloxacin exposure induced not only an increase in ROS levels and MDA content but also physiological cellular changes in Arabidopsis thaliana [96]. Riaz et al. (2017) indicated that exposure to ciprofloxacin, enrofloxacin, or levofloxacin or their mixture reduced wheat (*Triticum aestivum*) growth by causing oxidative stress during germination in a greenhouse sand culture study [95]. Otherwise, the net assimilation rate was inhibited, and the contents of photosynthetic pigments, chlorophylls, and carotenoids were reduced in wheat after exposure to ciprofloxacin [101]. Ciprofloxacin promoted changes in the carbon and nitrogen metabolism in plants by decreasing photosynthesis and nitrogenase activity, altering the plants’ amino acid profile and decreasing cell N concentrations [97]. Hillis et al. (2011) studied the effect of 10 antibiotics on seed germination and root elongation in three plant species: lettuce (*Lactuca sativa*), alfalfa (*Medicago sativa*), and carrot (*Daucus carota*). They found that plant germination was insensitive to the treatment concentration of 10,000 lg/L of chlortetracycline, levofloxacin, and sulfamethoxazole [98]. However, root elongation was the most sensitive compared with shoot and total length measurements. Decreases in leaf chlorophyll content were observed in yellow lupin seedlings and decreased its concentration by 47.7% and 48.5%, respectively, after exposure to ciprofloxacin and tetracycline in soil concentrations. Hu et al. (2021) found that the chemical speciation of ciprofloxacin in aqueous solution affected its phytotoxicity [100]. For example, ciprofloxacin exposure aggravated the growth inhibition of rice and induced oxidative damage responses when in a high pH aqueous solution. Thus, the use of FQ antibiotic-contaminated water for plant culture is a crucial concern.

## 4. Environmental Toxicology Mechanisms of FQs

### 4.1. Oxidative Stress-Related Mechanism

The main proven environmental toxicology mechanisms for FQ antibiotics include free radical production, DNA breaks, mitochondrial damage, apoptosis, and oxidative stress, as shown in Table 3. The physiological effects induced by oxidative stress on biological organization were demonstrated in vitro and in vivo. Oxidative stress is a consequence of interactions with various ROS-inducing agents existing in various niches. The production of ROS and oxygen-based radicals is constantly triggered by oxidative stress, and then the cellular antioxidants balance the production and repair of cellular damage [102]. Mutating genomes facilitate bacterial evolution by ROS. Aerobic bacteria maintain defense mechanisms against oxidative stress throughout their evolution. Bactericidal antibiotics kill by modulating their respective targets. This traditional view was challenged by studies that proposed an alternative, unified mechanism of killing, whereby toxic ROS were produced in the presence of antibiotics [103]. Chetelat et al. (1996) confirmed that the formation of active oxygen species appeared to be responsible for causing light-induced adverse effects [104]. Ray et al. (2006) found that the formation of reactive oxygen species by photoexcited drugs (lomefloxacin, norfloxacin, ofloxacin, and enoxacin) might be considered a possible mechanism of radiation-induced in vitro phototoxicity [105]. Spratt et al. (1999) reported that FQs can photochemically produce DNA damage by both type I (radical) and type II (singlet oxygen) mechanisms [106]. Gurbay et al. (2006) indicated that oxidative stress related to the DNA damage of ciprofloxacin induced in primary culture of rat astrocytes [107]. Pan et al. (2016) provided direct evidence that the increased risk of cellular oxidative stress by two FQs was closely related to the conformational and functional changes in the copper/zinc superoxide dismutase (Cu/Zn-SOD) molecule [108]. Beberok et al. (2010) indicated that the toxic effects of FQ antibiotics may also be caused by free radicals participating in the formation of these drug–melanin complexes [109]. Trisciuoglio et al. (2002) suggested that FQ antibiotic exposure to the cell membrane induced the lipid peroxidation chain reaction, in turn triggering ROS production. Their findings revealed that oxidative stress networks were the main targets for antimicrobial potentiation [110]. Huang et al. (2023) suggested that enrofloxacin-induced reproductive toxicity was related to germ cell apoptosis under oxidative stress in *Caenorhabditis elegans* [111]. This study suggests that chronic exposure to FQs at certain levels in the environment induces reproductive toxicity in nematodes and might reduce soil sustainability. High-dose enrofloxacin inhibited the growth of juvenile shrimp, caused gill and liver damage, and induced apoptosis of hepatopancreatic cells. These adverse effects were possibly caused by enrofloxacin-induced oxidative stress [112]. Norfloxacin nicotinate exposure to zebrafish larvae also increased the activities of SOD, CAT, and Gpx; increased the contents of MDA; and regulated gene levels related to antioxidant enzymes, which triggered oxidative stress [113]. Luo et al. (2018) indicated that lomefloxacin induced acute phototoxicity in Daphnia magna when exposed to UV radiation under oxidative stress [114].

ROS remain essential for FQ antibiotic stress in plants. Jin et al. (2022) found that the antioxidant response activated by ROS was the main effect of FQ antibiotic exposure, which triggered photosynthesis inhibition and cellular damage and was also an important toxicity mechanism [96]. After long-term exposure to two representative algae, *Prorocentrum lima* and *Chlorella* sp., norfioxacin increased antioxidant enzyme activities, increased ROS levels, unbalanced antioxidant systems, and played a positive role in the bloom of dinoflagellate red tides [123]. Hong et al. (2022) indicated that enrofloxacin (10 and 50 mg/L) inhibited the uptake of nitrogen and phosphorus by *Myriophyllum verticillatum*, triggered oxidative stress in leaves, and caused irreversible cellular damage [124]. The pro-oxidant effects induced by ciprofloxacin were evaluated by measuring oxidative stress biomarkers, such as catalase activity, and by determining lipoperoxidation levels. Nunes et al. (2019) found that ciprofloxacin significantly increased the activity of the antioxidant enzyme catalase and decreased the lipid peroxidation levels in *Lemna gibba* [115]. The data showed that the pro-oxidant character of ciprofloxacin activated the antioxidant defensive system of *Lemna gibba*. Oxidative stress can be an important contributor to the lethal effect of antibiotics in organisms. Thus, despite the different target-specific actions of bactericidal antibiotics, they have a common mechanism leading to organism self-destruction by the internal production of hydroxyl radicals and ROS. With the increasing use of FQ antibiotics, their ecological impact has attracted attention. The oxidative stress-related mechanism of FQ antibiotics is summarized in this part, as shown in Figure 2. However, research on the toxicity-related mechanism of FQ antibiotics also needs a systematic supplement. Additionally, some relevant pathways to target the oxidative stress response have been investigated in recent years, such as gene expression, DNA repair, and hydroxyl radicals. Furthermore, a deeper mechanism should be explored to determine the response to FQ antibiotic toxicity.

### 4.2. Nonoxidative Stress-Related Mechanism

Identifying the pathogens with decreased susceptibility to FQ antibacterial activity and revealing the molecular mechanism of resistance have important epidemiological and clinical implications in animal husbandry. Chen et al. (2011) found that nalidixic acid and enrofloxacin induced bacterial resistance in *Haemophilus parasuis* mainly caused by GyrA mutations [116]. Bai et al. (2012) discussed that the SOS response, target gene mutations, and efflux pump activity were the main mechanisms of Escherichia coli adaptation to enrofloxacin stress [117]. They also emphasized the distinction between the inherited and noninherited mechanisms in the adaptation of bacterial resistance. Sheng et al. (2008) found that the caspase-8-dependent mitochondrial pathway was the main mechanism by which ofloxacin induced apoptosis in microencapsulated juvenile rabbit chondrocytes [118]. Kocak et al. (2021) revealed that ribosomal processes, energy pathways (the tricarboxylic acid cycle and glycolysis), membrane proteins, microbial targets, and biofilm formation were different to the Escherichia coli resistance mechanisms of ofloxacin [125]. Soto et al. (2003) used in vitro FQ-resistant mutants of Salmonella enterica serotype Enteritidis to analyze the mechanisms involved in resistance [119]. They found that AcrAB-like efflux pumps played an important role in the increase in minimal inhibitory concentrations to some FQs. Regarding the antibacterial effect of FQs, Volff et al. (1994) showed that ciprofloxacin, enoxacin, and norfloxacin greatly stimulated genetic instability and the occurrence of DNA rearrangements in *Streptomyces ambofaciens* [126]. They suggested that the genotoxic effect observed could be due to an interaction with DNA gyrase. Mutations in the target genes in the quinolone resistance-determining region are important mechanisms related to ciprofloxacin resistance [127]. Mandell et al. (2001) used five mechanisms to explain adverse drug reactions (antimicrobial toxicity) with FQs, such as direct effects, hypersensitivity, changes in microbial flora, drug interactions, and microbial lysis [128]. The antibacterial mechanism of FQs involves disruption of the catalytic mechanism of topoisomerase IV and DNA gyrase in bacteria. For example, FQs damage enzymes that ligate cleaved DNA and damage the structure of the DNA [120]. Jiang et al. (2022) reported that the potential mechanism of mitochondrial protein synthesis was the main reason for mitochondrial toxicity induction after FQ exposure [129]. Some targets for FQ antibiotics in human cells have been found, such as FQs with DNA damage repair in eukaryotes [130]. Sendzik et al. (2005) found that, in addition to changes in receptor and signaling proteins, apoptosis must be considered a final event in the pathogenesis of fluoroquinolone-induced tendopathies in cultured human tendon cells [131]. Metterlein et al. (2011) found that ciprofloxacin or levofloxacin induced myotoxicity, which had a pathological influence on intracellular calcium handling [132]. Zhan et al. (2021) suggested that the main mechanism of virulence attenuation after the formation of resistance in an FQ-resistant strain (Pm64) was the regulation of the expression level of the involved genes [133]. Guo et al. (2020) suggested that enrofloxacin-treated hepatocytes were damaged via apoptotic signaling pathways and that composite ammonium glycyrrhizin prevented enrofloxacin-induced hepatocyte injury in chickens [134]. Ramon et al. (1999) studied the mechanisms of action of ciprofloxacin and norfloxacin bactericidal activity against staphylococci in vitro [121]. They found three mechanisms of in vitro bactericidal activity. The first was that it required cell division, as well as bacterial protein and RNA synthesis, to kill bacteria. The second was that it was active against nondividing bacteria but required protein and RNA synthesis. The last mechanism was that it was active against nondividing bacteria and did not require protein and RNA synthesis.

Shen et al. (2019) found that gatifloxacin induced morphological and functional abnormalities in the cardiovascular system of zebrafish [122]. Ciprofloxacin damaged the function of the cardiovascular system without morphological abnormalities. The downregulation of genes related to the calcium signaling pathway and cardiac muscle contraction may be the related cardiovascular toxicity mechanism. Currently, the coexistence of different concentrations of residual FQ antibiotics has significantly increased; specifically, the hazards of enrofloxacin and ciprofloxacin have drawn more attention. Mixing with ciprofloxacin increased the residual concentration of enrofloxacin in pigs, which changed the pharmacokinetics of enrofloxacin by inhibiting CYP3A29 binding to enrofloxacin [135]. Toxicity mechanisms are necessary to understand the toxicity effect of FQ antibiotics on the ecological system and organisms. Knowledge of toxic mechanisms is important to produce safe FQ antibiotic drugs for organisms and human health.

## 5. Toxicity Evaluation of Mixing Residual FQ Antibiotics and Other Pollutants

With FQ antibiotics being released into natural and man-made environments, many different kinds of pollutants are mixed together. As shown in Table 4, FQ antibiotics may experience physical chemical reactions with other pollutants that may change their physicochemical characteristics and affect biological toxicity. Some studies investigated the toxic effects of coexposure to FQ antibiotics in the environment. More attention was given to the toxicity of coexisting FQ antibiotics and other pollutants on organisms from environmental exposure. Therefore, understanding the joint toxicity effects of FQ antibiotics and their coexisting physicochemical characteristics is necessary to change the environmental compatibility and bioavailability of FQ antibiotic drugs. The combined toxicity effect of FQ antibiotics and compound contents showed antagonistic, additive, and synergistic effects. The mixed ratio significantly affected synergistic toxicity. For example, Luan et al. (2022) suggested that enrofloxacin–ciprofloxacin and enrofloxacin–florfenicol pairs exhibited synergistic cytotoxicity at specific concentrations in human hepatocytes, while enrofloxacin and sulfadimidine induced synergistic cytotoxicity at all tested concentrations [136]. Fan et al. (2022) found that enrofloxacin and carbendazim coexposure reduced the hatching rate at 48 h postfertilization and increased the hatching malformation and lethality rates at 96 h postfertilization in zebrafish embryos, which indicated that the antagonistic toxicity effects might be related to the reciprocal effects of metabolism-related genes, such as cyp7a1 and apoa1a [137]. Zhao et al. (2018) found that ciprofloxacin and zinc oxide (ZnO) nanoparticle complexes induced additive toxic effects on methanogenesis and the degradation of proteins and carbohydrates. ZnO nanoparticles+ciprofloxacin appeared to have complex toxicity effects on Firmicutes, Aminicenantes, Chloroflexi, and Parcubacteria [138]. Deng et al. (2021) studied the joint toxicities of hydrophobic pentachlorophenol, hydrophilic ciprofloxacin, and carbon nanotubes to Bacillus subtilis at the cellular, biochemical, and omics levels [139]. They found that carbon nanotube–pentachlorophenol and carbon nanotube–ciprofloxacin coexposure showed distinct additive and synergistic toxicities, respectively. Carbon nanotubes increased the bacterial bioaccumulation of pentachlorophenol and ciprofloxacin by destabilizing and damaging the cell membranes. Pentachlorophenol reduced the bioaccumulation of carbon nanotubes, while ciprofloxacin had no significant effect, which was due to the different effects of the coexisting organic contaminants on cell-surface hydrophobicity and carbon nanotube electronegativity.

The combined toxicity of FQ antibiotics and their coexisting chemicals in the environment includes nanoplastics, antibiotics, heavy metals, organic contaminants, and microplastics. For example, Guo et al. (2021) found that the presence of polystyrene nanoplastics reduced the toxic effects of ciprofloxacin on digestive glands and enhanced the inhibition rate of *Corbicula fluminea* siphoning [140]. However, the ciprofloxacin toxicity to *Corbicula fluminea* decreased due to the adsorptive action of micro- or nanopolystyrene on dissolved ciprofloxacin. Yang et al. (2022) suggested that greater comprehensive toxicity to soil enzyme activity (urease, sucrase, phosphatase, and Rubisco) was observed in combined enrofloxacin and copper treatment groups [141]. Ren et al. (2021) found that ciprofloxacin, norfloxacin, enrofloxacin, lomefloxacin, and their binary combinations caused synergistic toxicity on algae [142]. In addition, the joint toxicity could be decreased by changing the acid–base condition of the water. Zhao et al. (2019) revealed that norfloxacin and sulfamethazine (500 mg/kg) coexisting in sludge digestion reduced the methane production rate [143]. Nanoparticle–ZnO coexposure with norfloxacin induced the inhibition of hydrolysis, fermentation, and methanogenesis in different digestion periods. Yang et al. (2020) suggested that combinations of tetracycline–ciprofloxacin and tetracycline–norfloxacin reduced plant root elongation more than those of ciprofloxacin–norfloxacin [144]. The interaction of FQ antibiotics and other pollutants alters the ecotoxicity of individual FQ antibiotics. Antibiotics mainly migrate in the environment through adsorption into other environmental pollutants. The adsorption capacity of antibiotics and environmental pollution is closely related to their own structural properties. FQ antibiotics have many polar functional groups, such as carboxyl groups and hydroxyl groups, which easily adsorb particles in the environment. The adsorption capacity of antibiotics is also affected by various environmental factors. For example, different pH values can change the charge state of antibiotics and environmental media and significantly affect the adsorption capacity of antibiotics on pollution particles [145]. This part provides an overview of the potential ecological risks associated with the presence of FQ antibiotics and other pollutions in the environment. The environmental occurrence, transport, organism distribution, toxicity mechanism, and toxicity effect of FQ antibiotics are shown in Figure 3. Although the adsorption action from FQ antibiotics is the main factor of combined toxicity, a few studies analyzed the influencing mechanism of combined toxicities. Research on FQ antibiotics should fill the gap regarding combined toxicities in the ecological environment.

**Table 4 antibiotics-12-01058-t004:** The toxic effects of mixing FQ antibiotics and other pollutants.

Pollution	Species	Toxic Effect	References
Enrofloxacin combined with two antibiotics (ciprofloxacin and florfenicol)	Human hepatocytes	Synergistic cytotoxicity	[136]
Enrofloxacin combined with carbendazim	Zebrafish	Reduced the hatching rate and increased the hatching malformation and lethality rates	[137]
Ciprofloxacin combined with ZnO nanoparticle	Firmicutes, Aminicenantes, Chloroflexi, and Parcubacteria	Additive toxic effects on methanogenesis and the degradation of proteins and carbohydrates	[138]
Carbon nanotube combined with two antibiotics (pentachlorophenol and ciprofloxacin)	Bacteria	Additive and synergistic toxicities	[139]
Polystyrene nanoplastics combined with ciprofloxacin	Corbicula fluminea	Reduced the toxic effects of ciprofloxacin on digestive glands and enhanced the inhibition rate of Corbicula fluminea siphoning	[140]
Enrofloxacin combined with copper	Bacteria	Comprehensive toxicity to soil enzyme activity (urease, sucrase, phosphatase, and Rubisco)	[141]
Norfloxacin combined with sulfamethazine	Bacteria	Reduced the methane production rate	[143]
Tetracycline combined with two antibiotics (ciprofloxacin and norfloxacin)	Plant	Reduced plant root elongation	[144]

## 6. Conclusions

In this paper, the toxicity effects of FQ antibiotics on mammals, aquatic lives, bacteria, cells, and higher plants are summarized. The ecotoxicity of FQ antibiotics is well-studied in nontarget aquatic organisms. However, few toxicity studies on nontarget plants or sediment and soil-inhabiting nontarget organisms have been performed. Data on the low concentrations of FQ antibiotics in nontarget organisms and lifetime exposure experiments are lacking. A multidisciplinary approach is needed to reduce the long-lasting impacts of FQ antibiotics on sensitive ecosystems. The available data on the source and fate of FQ antibiotics in the ecological environment reveal that the toxicity effect of residues of FQ antibiotics on environmental organisms is highly dependent on many factors, including other residual pollutants, temperature, dissolved oxygen, and chemical properties. However, various environmental factors also cause the spread of antibiotic resistance. The wide use of FQ antibiotics has increased the transfer potential of resistance genes in species. Only some drug-resistant bacteria are recognized, but there is little knowledge about the process of natural transformation after exposure to antibiotics. The available data on the fate and effect of FQ antibiotics in the environment indicate that the prudent use of FQ antibiotics in humans as well as in veterinary medicine is imperative due to the worldwide antibiotic resistance crisis. Moreover, the toxic effects of multiple antibiotics and their mixed effect with other contaminants should be further studied. Information on the toxicity mechanism of single and joint FQ antibiotics will help in the assessment of toxicity effect relationships and the design of safe products by using risk assessment models. A systematic analysis of the relationships between the concentration effect of FQ antibiotics and other pollutants is necessary for assessing the negative disturbance of open environment ecosystems. Some advice is proposed for further research directions:Formulate an ecological toxicology evaluation system and guidelines to analyze the biotoxicity of FQ antibiotics at different concentration levels in multiple environments. Such a system will complete the prudent use guidelines.The standardized research methodology of FQ antibiotics is important for guiding enterprises in the design and production of drugs with high environmental biocompatibility.More information on the toxicity mechanism of FQ antibiotics is needed, which is essential for the government to control the discharge of pollutants into the environment and formulate policy coordination.

## Figures and Tables

**Figure 1 antibiotics-12-01058-f001:**
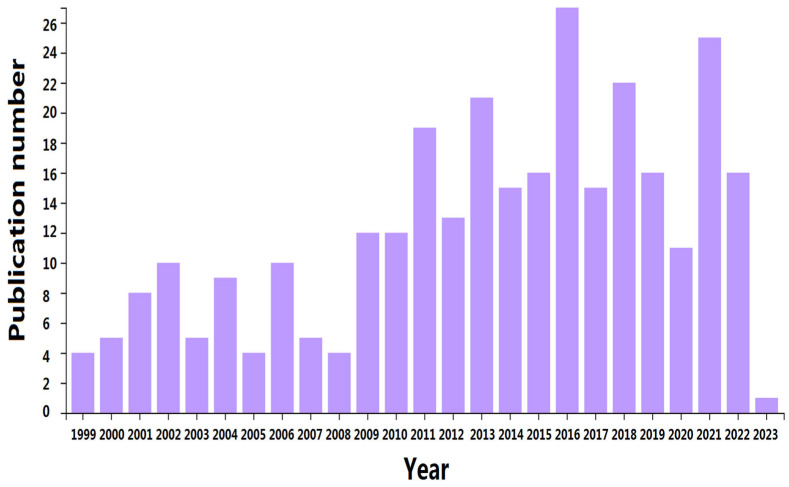
Number of publications on the toxicity of FQ antibiotics.

**Figure 2 antibiotics-12-01058-f002:**
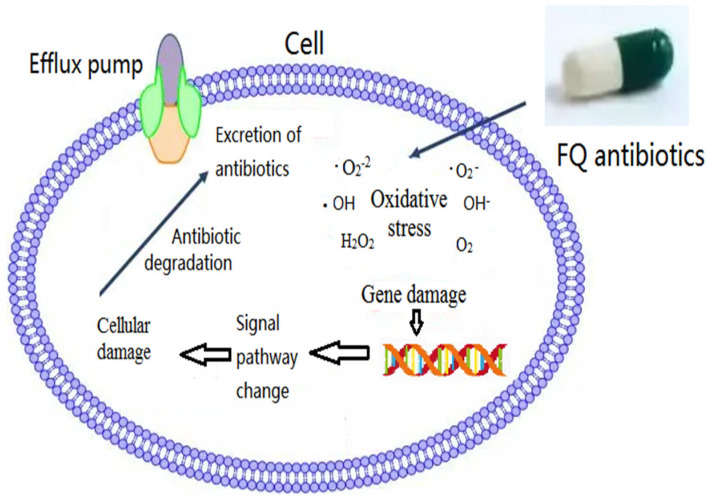
The oxidative stress−related mechanism of FQ antibiotics.

**Figure 3 antibiotics-12-01058-f003:**
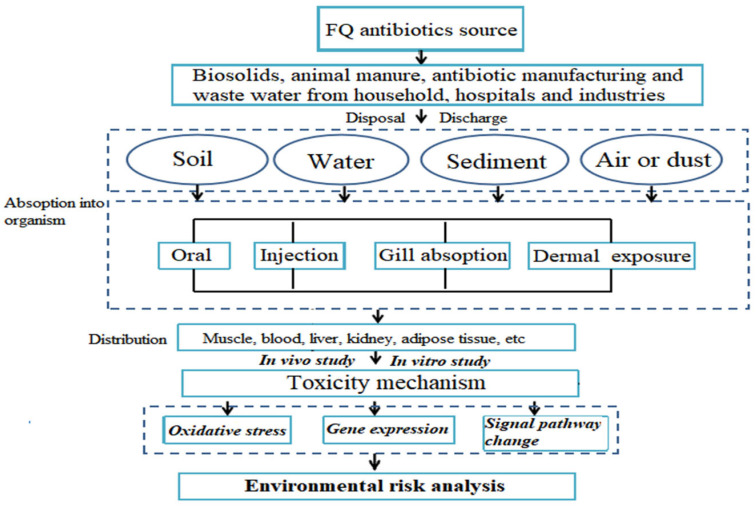
A model of FQ antibiotic action in toxicology.

**Table 1 antibiotics-12-01058-t001:** In vitro studies of FQ antibiotics on typical cells or bacteria.

Species	Types	Effect	References
HeLa cells	Norfloxacin, ciprofloxacin, and enoxacin	Phototoxicity, cytotoxicity, and DNA strand-breaking activity	[56]
Primary monkey hepatocytes	Moxifloxacin and gatifloxacin	Inhibiting fructose 1,6-bisphosphatase	[58]
Tendon cells (human, dog, mini pig, rat, marmoset)	Ciprofloxacin	Cytotoxicity and achilles tendon rupture	[59]
Corneal epithelial cell	Moxifloxacin and ciprofloxacin	Upregulation of type III secretion system and cytotoxic effects	[63]
Human corneal keratocytes and endothelial cells	Ciprofloxacin, gatifloxacin, levofloxacin, moxifloxacin, and ofloxacin	Cytotoxicity	[60]
Human corneal epithelial cells	Ofloxacin, levofloxacin, tosufloxacin, moxifloxacin, and gatifloxacin	Severe cellular morphological damage	[67]
Myosin and chicken meat proteins	Ciprofloxacin and sarafloxacin	Effect on meat proteins’ nutritional and technological properties	[61]
Streptococcus pneumoniae, Staphylococcus aureus, Staphylococcus epidermidis, and Haemophilus influenza	Besifloxacin, gatifloxacin, and moxifloxacin	Bactericidal activity	[68]
Intracellular bacteria of amoebae	Ofloxacin and ciprofloxacin	Antibacterial activity	[69]
Mycobacterium tuberculosis	Ciprofloxacin, ofloxacin, levofloxacin, sparfloxacin, and gatifloxacin	Antibacterial activity	[70]
Human retinal pigment epithelial cells	Ciprofloxacin and tetracycline	Upregulation of genes related to apoptosis, inflammation, and antioxidant pathways	[66]

**Table 2 antibiotics-12-01058-t002:** In vivo toxic effects of FQ antibiotics on species.

Species	Types	Effect	References
American shad	Enrofloxacin	Affects many signaling pathways, such as the glycolysis/gluconeogenesis and pentose phosphate pathways	[92]
Human	Moxifloxacin	Inhibition of bacterial enzymes needed for bacterial DNA synthesis	[94]
Mouse	Enrofloxacin, enoxacin, trovafloxacin, norfloxacin, and ofloxacin	Antibabesial effect	[81]
Zebrafish	Levofloxacin	Decreased mortality	[91]
Chicken	FQ antibiotics	Decreasing the morbidity and mortality associated with the treatment of antibiotic-resistant Campylobacter	[84]
Puffer fish	FQ antibiotics	Residues in dorsal-epaxial muscle of living puffer fish	[90]
Dog	Enrofloxacin, ofloxacin, orbifloxacin, and ciprofloxacin	Lack of inhibitory effects of several FQs on cytochrome P-450 3A activities	[85]
Rabbits	Moxifloxacin and gemifloxacin	Influence blood glucose levels and serum insulin levels	[87]
Rat	Ciprofloxacin and moxifloxacin	Reduced the hydrophobicity index of the bile acid pool, reduced secondary bile acids, and increased taurine-conjugated primary bile acids	[88]
Foal	Enrofloxacin	Severe cartilage erosions, synovitis, and joint capsular thickening	[89]
Pig	Enrofloxacin	Induced catalase (CAT) and glutathione peroxidase (Gpx) and increased CYP450 content in pig liver microsomes	[86]
Wheat	Ciprofloxacin, enrofloxacin, and levofloxacin	Damaged the physiological structure, reduced crop productivity, and decreased growth	[95]
Arabidopsis thaliana	Enrofloxacin, norfloxacin, and levofloxacin	Induced oxidative stress and increased reactive oxygen species (ROS) levels and malondialdehyde (MDA) content	[96]
Azolla	Ciprofloxacin	Decreased photosynthesis and nitrogenase activity and altered plants’ amino acid profile, with decreases in cell N concentrations	[97]
Lettuce, alfalfa, and carrot	Levofloxacin	Phytotoxic	[98]
Lupin	Ciprofloxacin	Decreased leaf chlorophyll content	[99]
Rice	Ciprofloxacin	Inhibited plant growth, decreased photosynthetic pigment contents, and enhanced antioxidant enzyme activities	[100]

**Table 3 antibiotics-12-01058-t003:** The main toxicology mechanisms for FQ antibiotics.

Toxic Effect	Mechanisms	References
In vitro phototoxicity	Formation of reactive oxygen species	[105]
DNA damage	Both type I (radical) and type II (singlet oxygen) mechanisms	[106]
DNA damage	Oxidative stress	[107]
Conformational and functional changes in Cu/Zn-SOD molecule	Cellular oxidative stress	[108]
ROS production	Lipid peroxidation chain reaction	[110]
Reproductive toxicity and cell apoptosis	Oxidative stress	[111]
Increase in the activity of the antioxidant enzyme catalase and decrease in lipid peroxidation levels	Oxidative stress	[115]
Bacterial resistance	GyrA mutations	[116]
Escherichia coli sensitivity to enrofloxacin	Target gene mutations and efflux pump activity	[117]
Apoptosis	Caspase-8-dependent mitochondrial pathway	[118]
Increase in minimal inhibitory concentrations to some FQs	AcrAB-like efflux pumps	[119]
Damage in the enzymes that ligate cleaved DNA and in the structure of DNA	Catalytic mechanism of topoisomerase IV and DNA gyrase	[120]
Bactericidal activity	Cell division as well as bacterial protein and RNA synthesis	[121]
Cardiovascular toxicity	Downregulation of genes associated with calcium signaling pathway and cardiac muscle contraction	[122]

## Data Availability

Data sharing not applicable.

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
