# Peer review of "The Research Status, Potential Hazards and Toxicological Mechanisms of Fluoroquinolone Antibiotics in the Environment"

_antibiotics, 2023, doi:10.3390/antibiotics12061058_

Round 1

Reviewer 1 Report

The topic of the article „The research status, potential hazards and toxicological mechanisms of fluoroquinolone antibiotics in the environment” is interesting, The Authors described the sources and fate of FQ antibiotics (FQs) in the environment as well as the in vitro and in vivo toxicity of the group of antibiotic including a proposed mechanism of the toxic effect. The manuscript requires editorial correction. Moreover, I have two minor specific comments:

1.       The sentence „Antibiotics are widely used in consumer products, such as anti-inflammatory 30 drugs, lipid regulators, antiepileptics, preservatives and sunscreen UV filters” (ln.30-31) is unclear and should be rewritten.

2.       Figure 1 – the font should be larger

Minor editing of the English language is required.

Author Response

1.The sentence„ Antibiotics are widely used in consumer products, such as anti-inflammatory drugs, lipid regulators, antiepileptics, preservatives and sunscreen UV filters” (ln.30-31) is unclear and should be rewritten.

Reply: This sentence was not properly expressed and has been rewritten in lines 39-41 as follows:

Antibiotics are widely used in medicines for treating bacterial infections in both people and animals due to their biologically active antibacterial, antifungal, and antiparasitic activities.

2. Figure 1 – the font should be larger

Reply: The font in Figure 1 was revised.

3. Comments on the Quality of English Language.Minor editing of the English language is required.

Reply: We have revised the language using an editing service.

Reviewer 2 Report

Dear Authors,

 In this paper the authors show the toxicity effects of FQ antibiotics on mammals, aquatic lives, bacteria, cells and higher plants which are summarized. Available data on the source and fate of FQ antibiotics in the ecological environment reveal that the toxicity effect of residues of FQ antibiotics on environmental organisms is highly dependent on many factors, including other residual pollutants, temperature, dissolved oxygen and chemical properties. Systematic analysis of the relationships between the concentration effect of FQ antibiotics and other pollutants is necessary for assessing the negative disturbance of open environment ecosystems.

The description of the work is acceptable. Overall impression is that this manuscript can be recommended for publication after MAJOR revision in Antibiotics, especially considering the scope and topics of this journal. However, I would like to point out to several details:

  1. Underscore the scientific value added of your paper in your abstract. Please look at articles we have published for models. Your abstract should clearly state the essence of the problem you are addressing, what you did and what you found and recommend. That will help a prospective reader of the abstract to decide if they wish to read the entire article. This should be corrected.
  2. Please note in the Instruction of the Microorganisms for Introduction and state the objectives of the work and provide an adequate background, avoiding a detailed literature survey or a summary of the results. Correct this.
  3. It is not clear what novelty in paper worth to publish is? Correct this.
  4. In the conclusions, in addition to summarizing the actions taken and results, please strengthen the explanation of their significance. It is recommended to use quantitative reasoning comparing with appropriate benchmarks, especially those stemming from previous work. This should be corrected.
  5. Figure 1. and 2. should not be in the conclusion section. Correct this.
  6. For all section in the discussion the authors should give the table for the summarizing the paper results which they are analyze. Correct this.

7.      English language should be corrected by a professional lector. A proof reading by a native English speaker should be conducted to improve both language and organization quality.

 I wish a lot of success to the authors in making this manuscript much better.

With kind regards!

Reviewer 

English language should be corrected by a professional lector. A proof reading by a native English speaker should be conducted to improve both language and organization quality

Author Response

1.Underscore the scientific value added of your paper in your abstract. Please look at articles we have published for models. Your abstract should clearly state the essence of the problem you are addressing, what you did and what you found and recommend. That will help a prospective reader of the abstract to decide if they wish to read the entire article. This should be corrected. 

Reply: The abstract has been rewritten in the text as follows:

Fluoroquinolone antibiotics are widely used in pharmaceuticals and personal care products due to their disease treatment, growth promotion, and prophylaxis, which lead to increasing biological exposure to fluoroquinolones in the global environment. This paper recapitulates the occurrence, fate, and ecotoxicity of fluoroquinolone antibiotics in various environmental media. The toxicity effect is reviewed based on in vitro and in vivo experiments, referring to many organisms, such as microorganisms, cells, higher plants, land and aquatic animals. Furthermore, a comparison of various toxicology mechanisms of fluoroquinolone antibiotic residues on environmental organisms is made. This study identifies gaps in the investigation of the toxic effects of fluoroquinolone antibiotics and the mixture of multiple fluoroquinolone antibiotics on target and nontarget organisms. The study of the process of natural transformation toward drug-resistant bacteria is also recognized as a knowledge gap. The review also details the combined toxicity effect and the adsorption capacity on other chemicals to organisms in various environmental matrices, and the scarcity of data on the ecological toxicology evaluation system of fluoroquinolone antibiotics are identified. The present study entails a critical review of the literature providing guidelines for the government to control the discharge of pollutants into the environment and formulate policy coordination. Future study work should focus on developing a standardized research methodology for fluoroquinolone antibiotics to guide enterprises in the design and production of drugs with high environmental biocompatibility .

2.Please note in the Instruction of the Microorganisms for Introduction and state the objectives of the work and provide an adequate background, avoiding a detailed literature survey or a summary of the results. Correct this.

Reply: This part was revised from lines 225 to 259. The objectives of the work and an adequate background have been added. The detailed literature survey or a summary of the results has been removed.

3.It is not clear what novelty in paper worth to publish is? Correct this.

Reply: The innovation of the article is mentioned in the abstract.

This study identifies gaps in the investigation of the toxic effects of fluoroquinolone antibiotics and the mixture of multiple fluoroquinolone antibiotics on target and nontarget organisms. The study of the process of natural transformation toward drug-resistant bacteria is also recognized as a knowledge gap. The review also details the combined toxicity effect and the adsorption capacity on other chemicals to organisms in various environmental matrices, and the scarcity of data on the ecological toxicology evaluation system of fluoroquinolone antibiotics are identified. The present study entails a critical review of the literature providing guidelines for the government to control the discharge of pollutants into the environment and formulate policy coordination. Aside from providing an in-depth look at the development and risks of FQ antibiotics when they enter the environment, the present study entails a critical review of the literature and motivates researchers to carry out further research.

4. In the conclusions, in addition to summarizing the actions taken and results, please strengthen the explanation of their significance. It is recommended to use quantitative reasoning comparing with appropriate benchmarks, especially those stemming from previous work. This should be corrected.

Reply: The relevant content has been added to the text (lines 530-535 and lines 539-547).

5. Figure 1. and 2. should not be in the conclusion section. Correct this.

Reply: The original Figure 1 was added in lines 98. The original Figure 2 was added in line 454.

6.For all section in the discussion the authors should give the table for the summarizing the paper results which they are analyze. Correct this.

Reply: Table 3 and Table 4 have been added in the text.

7. English language should be corrected by a professional lector. A proof reading by a native English speaker should be conducted to improve both language and organization quality.

Reply: We have revised the language using an editing service.

Reviewer 3 Report

Jia et al in this review manuscript titled "The research status, potential hazards and toxicological mechanisms of fluoroquinolone antibiotics in the environment" have described the occurrence, fate, and ecotoxicity of fluoroquinolone antibiotics in various environmental media. 

The manuscript is well-written and scientifically sound. I suggest authors to include more number of figures. At least the mechanism of action of fluoroquinolone. This would enrich the work.

Author Response

Reply: A figure showing the mechanism of action of fluoroquinolone was added to the text (line 392).

Round 2

Reviewer 2 Report

Dear Authors,

In my opinion the paper is much better than earlier, and I recommend the publication this paper in present form in Antibiotics.

Best wishes,

Reviewer

Author Response

  • Dear reviewer
  •  
  • Thank you very much for the reviewer's careful correction. Subsequent modifications have been made in accordance with the editor's requirements.

Best wishes,

Authors